# An Insight into the Algal Evolution and Genomics

**DOI:** 10.3390/biom10111524

**Published:** 2020-11-06

**Authors:** Amna Komal Khan, Humera Kausar, Syyada Samra Jaferi, Samantha Drouet, Christophe Hano, Bilal Haider Abbasi, Sumaira Anjum

**Affiliations:** 1Department of Biotechnology, Kinnaird College for Women, Lahore 54000, Pakistan; aaykay28@gmail.com (A.K.K.); humera.kausar@kinnaird.edu.pk (H.K.); syyada.samra@kinnaird.edu.pk (S.S.J.); 2Laboratoire de Biologie des Ligneux et des Grandes Cultures (LBLGC), INRAE USC1328, Université d’Orléans, 28000 Chartres, France; samantha.drouet@univ-orleans.fr (S.D.); hano@univ-orleans.fr (C.H.); 3Department of Biotechnology, Quaid-i-Azam University, Islamabad 45320, Pakistan; bhabbasi@qau.edu.pk

**Keywords:** algae, bio-products, bioengineering, *Chlamydomonas reinhardtii*, evolution, genomics, lignin, phenylpropanoids, phylogenetics, phyla

## Abstract

With the increase in biotechnological, environmental, and nutraceutical importance of algae, about 100 whole genomic sequences of algae have been published, and this figure is expected to double in the coming years. The phenotypic and ecological diversity among algae hints at the range of functional capabilities encoded by algal genomes. In order to explore the biodiversity of algae and fully exploit their commercial potential, understanding their evolutionary, structural, functional, and developmental aspects at genomic level is a pre-requisite. So forth, the algal genomic analysis revealed us that algae evolved through endosymbiotic gene transfer, giving rise to around eight phyla. Amongst the diverse algal species, the unicellular green algae *Chlamydomonas reinhardtii* has attained the status of model organism as it is an ideal organism to elucidate the biological processes critical to plants and animals, as well as commercialized to produce range of bio-products. For this review, an overview of evolutionary process of algae through endosymbiosis in the light of genomics, as well as the phylogenomic, studies supporting the evolutionary process of algae was reviewed. Algal genomics not only helped us to understand the evolutionary history of algae but also may have an impact on our future by helping to create algae-based products and future biotechnological approaches.

## 1. Introduction

The application of genomic approaches, such as genome sequencing and gene function analysis, to algal research has caused a step change in our understanding of algal biology, ecology, and evolution. The rise of next-generation sequencing (NGS) technologies at a lower cost has led to the acquisition of significant genomic data on algae since 1990s [1]. Whole genomes of large-sized organisms can be sequenced more easily along with the availability of a greater number of bioinformatics tools for data assembly and annotation, easing-up the sequence analyses and prediction of the encoded biological function [2]. Genomics is a field concerned with defining sequences of genes and genomes and analysis of those sequences to understand the organization, architecture, expression, evolution, and function of genes and genomes [3]. With the increased biotechnological, environmental, and nutraceutical importance of algae, about 100 whole genomic sequences of algae have been published, and this figure is expected to double in the coming years [4].

Algae diversity range from tiny unicellular microalgae to giant seaweeds which can grow over 50 m long, abundantly found in virtually every ecosystem [5]. These ancient organisms are defined as unicellular or multicellular photosynthetic organisms regarded as polyphyletic due to morphological resemblance with plants but do not share a common ancestor [6]. Similar to plants, algae are autotrophs, photosynthesize, and produce the same storage compounds and defense strategies. On the contrary, algae do not possess specialized root, stem, nor vascular bundles, and they lack diploid embryo stage. The characteristic features of algae include autotrophic chlorophyll-bearing thalloid plant body, lacks sterile tissue around its reproductive structures, and zygote development is by mitosis or meiosis but not via embryo formation [7]. Algae are not only diverse in their size but also in their ecological distribution, cellular biology, photosynthetic pigments, structural and reserve polysaccharides, and evolutionary origin of heterogeneous algae include prokaryotic and eukaryotic species [8].

Algae are polyphyletic as their origin cannot be traced back to single common hypothetical ancestor. The symbiogenic events believed to have occurred more than 1.5 billion years ago were the reason algae came into existence, and, because of so much time gap, it is difficult to track the key events that resulted in algae occurrence [9]. It is thought that they came into existence when a photosynthetic cyanobacteria invaded a unicellular eukaryote giving rise to double-membranous primary plastid. This is known as primary endosymbiosis and as a result of this endosymbiosis Green algae (Chlorophytes and other land plants), Red algae (Rhodophytes), and Glaucophytes came into existence [10]. Another secondary endosymbiosis occurred when other heterotrophic eukaryotes engulfed red and green algae, due to which many other algal species came into existence, forming three or four membranes of secondary plastid [11]. The molecular sequence analysis suggests that there exist around eight to nine major phyla (divisions) of algae. These are the cyanobacteria (chloroxybacteria) and the eukaryotic phyla Dinophyta (dinoflagellates), Glaucophyta, (glaucophytes), Cryptophyta (cryptomonads), Euglenophyta (euglenoids), Ochrophyta (a diverse array of tiny flagellates, chrysophyceans, diatoms, brown algae, and many other groups), Haptophyta (haptophytes), Rhodophyta (red algae), and Chlorophyta (green algae) [12].

Algae are not only the primary producers in food chain and contributing enormously to various processes of ecosystem, they are also economically very important. Algae are a rich source of pharmaceutically important primary and secondary metabolites exhibiting anti-viral, anti-bacterial, and anti-cancerous activities [13,14]. Algae is also being exploited as an alternate renewable source of energy in the form of biofuels, bio-oils, biodiesel, bio-hydrogen, bio-methane, bio-ethanol, and bio-butanol [15]. Algae is a rich source of vitamin, pigments, carotenoids, fertilizer, fiber, agar, alginate, and numerous other products [16]. Interestingly, algae are consumed as food in the form of soups, and salads around the world. The famous algal-food wakame, nori, or wrap for sushi is rich in nutrition. Moreover, they are used as animal feed to improve animal health and quality of animal meat [17]. These are the uses of some of the algal species, while more than 30,000 algal species have already been described, and other algal groups still await discovery [18]. The phenotypic and ecological diversity among algae hints the range of functional capabilities encoded by algal genomes. In order to explore the biodiversity of algae and fully exploit their commercial potential, understanding their evolutionary, structural, functional, and developmental aspects at genomic level is a pre-requisite.

## 2. Insights to Algal Evolution

There is a general consensus that first photosynthetic eukaryote emerged from a heterotrophic eukaryote which captured a coccoid cyanobacterium about 1.6 billion years ago. Gradually, the cyanobacterium was enslaved, and the endosymbiont’s genes transferred to the host nucleus as a new organelle called as primary plastid. This event is named as primary endosymbiosis [19]. The genome of cyanobacteria underwent evolution, resulting in either loss or transfer of genes to the host nucleus in a process termed as endosymbiotic gene transfer (EGT). As a result, only little portion of genome retained within primary plastid, including some genes encoding for photosynthesis [20]. The primary endosymbiosis gave rise to three groups, having primary plastids, forming a major group of autotrophic eukaryotes Archaeplastida: Glaucophytes, Red algae, and Green algae. Genomes of primary plastids consist of highly conserved genes employed extensively to study the evolution of diverse Archaeplastida groups [21].

The Red algal lineage or Rhodophyta consist of 5000–6000 multicellular eukaryotic marine algal species. Phylum Rhodophyta is comprised of two sub-phyla Rhodophytina and Cyanidiophytina with seven classes [22]. Rhodophyta have evolved with plastids lacking chlorophyll accessory pigments, rather, containing chlorophyll-a, phycocyanin, and phycoerytherin, so light is captured by phycobiliproteins [23]. Secondly, there is complete absence of centrioles and flagella [24]. The genome in the red algae are compact in nature and have high gene capacity. The mitochondrial genome in Florideophyceae are considered highly conserved despite the loss of 32 genes, which suggest endosymbiotic gene transfer [25]. Genomic and transcriptomic analysis of four different classes of Red algae (Compsopogonophyceae, Stylonematophyceae, Rhodellophyceae, and Porphyridiophyceae) reveals significant evolutionary insights. The proteins from the plastid genome of 37 red algae were translated and analyzed, which supported the idea that all four classes show monophyly and are deeply rooted in red algal phylogenetic tree [26].

Green algae are termed as Chlorophyceae, and they are characterized by the presence of double membranous chloroplasts, stacked thylakoids, and photosynthetic pigments: chlorophyll-*a* and *b* [27]. Within their plastids they have starch molecules as polysaccharide reserve, a characteristic similar to green plants. The *Chlamydomonas reinhardtii* mutants showed starch biosynthesis similar to starch biosynthesized by maize endosperm [28]. Chlorophyceae contains five lineages: Chlamydomonadales, Chaetopeltidales, Oedogoniales, Sphaerpleales, and Chaetophorales [29]. They emerged about 470 million years ago, which marks one of the important evolutions of time, adapted all the photic zones of Earth, and paved way for the evolution of other life [30].

The glaucophytes (glaucocystophytes) are a group of freshwater microalgae which contain blue-green plastids often called as cyanelles. In five genera of glaucophytes, about thirteen species have been described. Glaucophyte plastids are similar to cyanobacteria, and different from plastids of other algae, in having a thin peptidoglycan wall and contain only phycobilins and chlorophyll *a* [31]. Glaucophyte is a poorly studied genera probably due to their rareness in nature and restricted habitat. Genomic data and a better understanding of the phylogenetic position of glaucophytes will provide valuable insights into the endosymbiotic origin and evolution of plastids in eukaryotes [32].

A phylogenomic study revealed a sister-group relationship between green or red algae and glaucophytes with a bootstrap value ≥ 90% [33]. The comparison of genome of diatom with other organism revealed that many genes present in its genome were obtained by the horizontal gene transfer (HGT) from bacteria. Moreover, viruses as gene transfer agents are of considerable importance in the marine environments [34]. Along with achieving new genes with HGT and EGT, gene loss, fusion and duplication have also added to size of algal genome. There is a great difference in the genome size among eukaryotes because of genome organization, like gene distribution. For example, *Ostreococcus tauri*, contain 8166 genes located on 20 chromosomes and spanning a genome size of 1.54 kb/gene [35].

Algae outside Archaeplastida have their plastids originating from acquisition of primary plastids during secondary endosymbiosis. This gave rise to haptophytes, cryptophytes, dinoflagellates, heterokonts, and other photosynthetic eukaryotes [36]. The existence of secondary endosymbiosis was first indicated by the occurrence of more than two envelope membranes around the secondary plastids. As evident from the presence of four membrane bounding plastids in chlorarachniophytes, cryptomonads, ochrophytes, and haptophytes, dinoflagellates and euglenoids appear to have lost one of these membranes possessing three [37]. Moreover, nucelomorph (remnant of endosymbiont nucleus) can be found within chlorarachniophytes and cryptophytes [38]. The process of algal evolution through endosymbiosis is depicted in Figure 1.

To add to the complexity of algal evolution, there seems to exist tertiary endosymbiosis too. Some dinoflagellates have gained their plastid either by replacing the red-alga-derived plastid with a green-alga-derived plastid, or by capturing secondary endosymbiotic cryptophyte, haptophyte or heterokont alga in tertiary endosymbiosis [39]. Genomic data is rapidly accumulating to help us further understand the evolutionary process of photosynthetic eukaryotes, which appears to be a complex and haphazard process.

## 3. Algal Genomes

The last decade has witnessed intense progress in algal genome sequencing with many whole-genome sequences (WGS) of representative algae available for major lineages. Green microalgae are single-celled green algae which inhabit a diverse ecosystem. The unicellular green algae *Chlamydomonas reinhardtii* has attained the status of model organism as it is ideal organism to elucidate the biological processes critical to plants and animals [40]. It is a haploid organism, consist of 110 Mb sized nuclear genome spanning over 17 chromosomes, a 16 kb mitochondrial genome, and chloroplast genome of 203 kb, encodes for 14,000 proteins [41]. Comparative genomic analysis of *Chlamydomonas* genome can help trace the green plant or plant-animal common ancestor. The BLASTp (Basic Local Alignment Search Tool for searching protein collections) comparison between *Chlamydomonas* proteome with human and *Arabidopsis* proteome revealed more similarity with *Arabidopsis*, whereas basal body and flagellar proteome showed similarity with human homologs [42].

Diatoms (Bacillariophyta) are found abundantly in every aquatic photic zone, emerging between 225–200 million years ago [43,44]. WGS of several representative diatoms have become available which includes *Thalassiosira oceanica*, *Thalassiosira pseudonana, Cyclotella cryptica, Phaeodactylum tricornutum, Fragilariopsis cylindrus, Fistulifera solaris, Pseudo-nitzschia multistriata*, and *Synedra acus* [45]. In the phylogeny-based HGT study on all sequenced diatoms genomes, 1979 genes were found to be of horizontal origin in diatoms. Furthermore, five vitamin B12 biosynthesis pathway genes were detected as HGT, imparting a competitive advantage to diatoms in cobalt depleted environment, as cobalt is essential co-factor for diatoms [46]. In an iron-scarce environment, diatoms have the ability to reduce their cellular iron requirement and replaces plastid-localized ferredoxin with flavodoxin, using flavin as co-factor. This feature is similar to cyanobacteria and most other algae [47,48]. *Osterococcus tauri* and related species are a phytoplankton, and their smallest eukaryotes, known having a diameter of ≈ 1μM, play a significant role in global primary productivity, food chain, and biogeochemical cycles [49]. The ability of *O. tauri* to thrive oceanic environment of low iron availability is a good model to study genetic adaptation of algae in ocean environment. The RNA sequencing of *O. tauri* under iron limitation showed that iron uptake and metabolism are tightly coordinated with diurnal cycles and differ fundamentally from *C. reinhardtii* iron metabolism [50].

The genome analysis also reveals insights into the genome mutations and its consequences of protein, as well as environmental, survival. Recently, transcriptome analysis of *C. reinhardtii* revealed cadmium tolerance genes and pathways on exposure to 0.5 μM cadmium for over 420 days. Genome analysis suggested a mutation and differential expression of plasma membrane calcium-transporting ATPase encoding gene (CHLREDRAFT_189266), calmodulin binding protein encoding gene (CSE1) and calcium ion binding genes (CHLREDRAFT_187187, CHLREDRAFT_191203) in mutant and wild-type strains [51]. Moreover, differential expression of genes associated with cell cycle, protein synthesis and protein kinase-based phosphorylation in *C. reinhardtii* was observed under cold stress [52]. Such studies are important in enhancing our understanding of evolution of algae under harsh environment and comparing with other organisms, like plant stress tolerance.

Genomes of algae inhabiting extreme environments helps us understand the adaptive evolution of extremophile algae from their neutrophilic ancestors. *Chlamydomonas eustigma* is an acidophilic green algae and the comparative genome analyses between *C. eustigma* and *C. reinhardtii* (neutrophilic relative) revealed higher expression of H^+^-ATPase and heat-shock proteins, as well as loss of metabolic pathways, that acidify cytosol by HGT in acidophile. Similar features are found in red and green acidophilic algae, hinting at a common evolutionary mechanism [53]. *Chlamydomonas sp.* ICE-L and *Tetrabaena socialis* are two Antarctic psychrophilic green algae thrive in extreme polar environment. Their phylogenetic analysis showed proofs of positive evolution and complex modifications in genes encoding for photosynthesis and protein synthesis. These adaptations include highly stable photosynthetic apparatus, enhanced membrane fluidity, antioxidant strategies, and proteins for cold response [54].

Cyanidiophytina red algae, sister to mesophilic red algae (the Rhodophytina), is a unicellular taxa able to thrive at pH 0–4 and temperatures extremes up to 56 °C. *Galdieria sulphuraria* is a hot-spring dwelling cryptoendolithic encompass 13.7 Mbp genomic sequence, demonstrated HGT from prokaryotic sources, and extensive gene loss in common ancestor [55].

The small genome of unicellular red algae *Cyanidioschyzon merolae* provides useful information on basic and essential genes required for the survival of autotrophic eukaryotes. The unique features of 20 chromosomal *C. merolae* genome were presence of very few introns, three copies of ribosomal DNA involved in the maintenance of nucleolus, unique telomeric repeats for chromosomal ends, and a small number of genes [56]. Moreover, the mosaic origin of Calvin cycle was identical in *C. merolae* and *A. thaliana*, supporting the single primary endosymbiotic plastid concept [57].

Studies based on comparative transcriptomic analysis of single genes also provide useful insights of algal evolution. Han and colleagues studied the origin and evolution of xylosyltransferases (XylTs), which are polysaccharide biosynthesis enzymes, in 22 sequenced primary endosymbionts (glaucophytes, red algae, and green algae), as well as in secondary endosymbionts (diatoms, brown algae, Eustigmatophyceae, Cryptophyta, and Pelagophyceae). The study revealed that XylTs are distributed in Choanoflagellatea and Echinodermata, suggesting their origin from last common eukaryotic ancestor instead of being plant-specific [58]. Similarly, GRAS (Gibberellic-acid insensitive (GAI), Repressor of GAI (RGA) and Scarecrow (SCR)) and PYR/PYL/RCAR (Pyrabactin Resistance1 (PYR1)/PYR1-like (PYL)/regulatory components of ABA receptors (RCAR)) are genes which help zygnematophyceae and embryophytes survive biotic and abiotic stress. The two *Mesotaenium endlicherianum* and *Spirogloea muscicola* zygnematophyceae species were analyzed for their genome, revealed the origin of stress genes through HGT from soil inhabiting bacteria in common ancestor of zygnematophyceae and embryophytes [59].

Studying plastid genome of algae reveals information about algal evolution through endosymbiosis. *Emiliania huxleyi* is a marine calcifying phytoplankton belonging to coccolithophorids [60]. Its plastid DNA is densely packed and smaller than other red plastid lineages. The comparison of gene content, gene cluster, and gene function with plastid genomes of primary plastid and secondary plastid containing algae was drawn. Results disclosed that *Emiliania huxleyi* is related more to chlorophyll *c*-containing cryptophytes and heterokonts, and supports the descent of chromophyte from red algae [61]. Plastid genome comparison of microalgae *Gracilaria chilensis* with 17 red algal plastid sequences suggested separation of red algae into different groups, where Florideophyceae formed a large branch and Gracilariales as its sub-branch [62]. *Helicosporidium* is a non-photosynthetic alga, in which the plastid genome is evolved to have lost genes for major metabolic functions. Comparison with green algae and other distantly related non-photosynthetic plastids shows possibility of convergent evolution with apicomplexa [63].

In-depth knowledge of algal genomes can provide a thorough insight into the evolution of metabolic pathways critical to understanding the processes of evolution in the green lineage. One good example is the phenylpropanoid pathway including the lignification process. About 475 million years ago, the aquatic ancestral terrestrial plants adapted to form lignified cell walls which is widely regarded as one of the key innovations in the process of evolution [64]. Lignin plays a pivotal role in transport of water and structural support of plants by strengthening the secondary cell walls of xylem tissues and creates a dense matrix. The precursors of lignin may also have antimicrobial properties and may offer protection when a plant is damaged [65]. Such developmentally specialized cell walls were thought to be restricted to vascular plants as adaptations to terrestrial habitats within the algal lineage. The discovery of lignin or its precursor in different algae led to ponder over the evolution of its biosynthetic pathway [66]. The genes encoding for enzymes involved in the synthesis of *p*-coumaryl alcohol, which is a simplest lignin monomer (H units) have been reported in a range of photosynthetic eukaryotes. Moreover, presence of different lignin units (i.e., G, S, and H units, Figure 2) have been found in the cell walls of calcified red alga *Calliarthron cheilosporioides* (Rhodophyta), as well as in other algae [67]. In algae, putative key genes involved in the biosynthesis pathway of this simple lignin precursor have been identified in the haptophyte *Emiliania huxleyi*. The presence of *p*-coumaric acid and flavonoids have been also reported in diatom (*Phaedactylum tricornutum*) [68]. The potential of diatoms and hapthophytes to biosynthesize monolignols suggests the possible evolution of this biosynthetic pathway in ancient oceans and later predating the origin of land plants, despite the fact that they are evolutionary distant from each other and plants, as well [67].

Interestingly, there are three main enzymes (cinnamoyl-CoA reductase (CCR), 4-coumarate:CoA ligase (4CL), and cinnamyl alcohol dehydrogenase (CAD)) involved in the biosynthesis pathway of *p*-coumaryl alcohol which are able to produce sufficient level of *p*-coumaryl alcohol from *p*-coumaric acid [67]. The phylogenetic analysis of the genes encoding these three enzymes reveals ancient origin of this metabolic pathway as their homologous enzymes are present in red algae, green algae, dinoflagellates, glaucophytes, cryptophytes, haptophytes, and diatoms alongside land plants. During the evolutionary process, these lignin biosynthesis enzymes evolved to become multifunctional or paralogues with divergent functions, such as a CAD enzyme, which catalyzes the last step in monolignol biosynthesis. Since red algae are polyphyletic, there might be some paralogs weakly clustering with homologs from heterotrophic eukaryotes or bacteria, whereas the *p*-coumaryl alcohol genes seem to be present in shared ancestor of green algae and plants. However, the presence of pathway in the archaeplastid ancestor itself is difficult to predict before glaucophytes diverged. Further genomic analysis reveals the EGT of *p*-coumaryl alcohol biosynthesis pathway from the ancestors of green or red algae or themselves [67]. Further studies on the lignin or monolignols would help in filling knowledge gaps to understanding the crucial roles of these metabolites in terrestrial adaptation in the green lineage.

## 4. Conclusions and Future Prospects

The rapid increase in the number of sequenced genomes of algal species is opening doors of knowledge for scientists and giving us a new perspective to understand these very diverse and complex photosynthetic eukaryotes. Through phylogenomics and whole genome sequencing, we are able to recognize functional potential of individual algae, their adaptability to each ecological niche, and the movement of these genes among different organism through span of evolution.

The simplicity of algal genome, fast growth, direct synthesis of specific products, and the ability to fix CO_2_ by capturing solar energy, algae serves an exceptional candidate as cellular factories for the production of value-added pharmaceuticals, nutraceuticals, and bioenergy. The molecular tools, such as sequencing technologies and genome editing, are empowering the research on algae and development of algal bioproducts [68]. Genetic modification of metabolic pathways in algae can help enhance yield of valuable products [69]. *C. reinhardtii* appears as one such algae which have a prominent role in basic research on algae that has allowed scientists to understand and genetically engineer algal metabolism. *C. reinhardtii* is among the first engineered algae to be studied at commercial scale; therefore, it helped to understand and improve the production of biofuels and bio-products in algae [69]. It was engineered for heterologous production of diterpenoids, which have health promoting benefits [70]. Algal genomics has also provided wealth of information regarding genes involved in lipid biosynthesis. This information has helped in strain engineering to enhance lipid content for biofuel development [71]. However, the naturally low oil content, weak growth in outdoor ponds and sensitivity to high solar irradiation are some of the limitations in this regard which can be overcome with genetic engineering [69] (Figure 3). Given the generally regarded as safe (GRAS) status of *C. reinhardtii*, it can also be used as an effective production system for recombinant proteins at a lower cost and in larger quantities than in other eukaryotic production models, such as Chinese Hamster Ovary (CHO) cells [69,72]. To date, different therapeutic recombinant proteins have been produced in algae with proper post-translation modifications, most of them produced in *C. reinhardtii*, such as the erythropoietin hormone [73,74], or the anti-cancerous TNF (tumor necrosis factor)-related apoptosis inducing ligand [75].

To summarize, it is clear that the algal genomics has not only helped us in understanding evolutionary history of algae, but it has impacted our future by helping in development of algal based products and future biotechnology approaches.

## Figures and Tables

**Figure 1 biomolecules-10-01524-f001:**
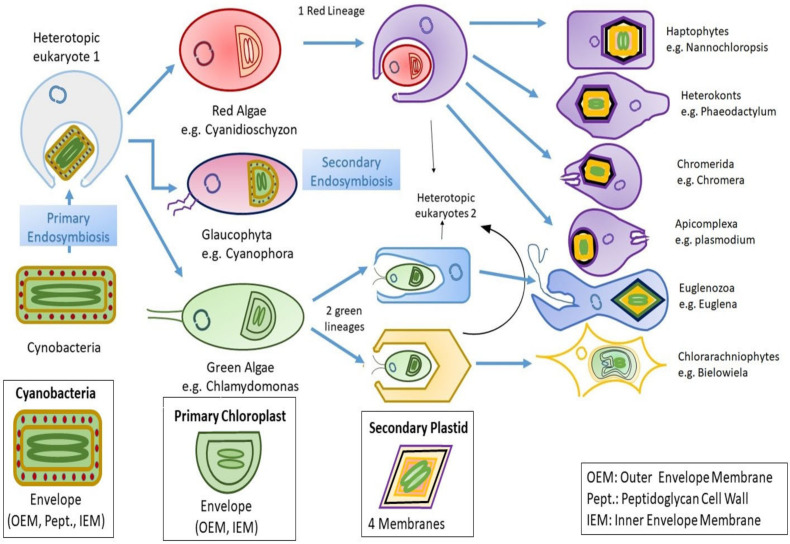
Diagrammatic representation of endosymbiosis: Primary endosymbiosis occurs when a heterotopic eukaryote engulfs a prokaryote (gram negative cyanobacteria), which leads to emergence of three lineages bearing primary chloroplast with two membranes that it vertically inherited from the cyanobacteria. Later on, the independent secondary endosymbiosis of green algae (eukaryote) by two unknown heterotopic eukaryotes that leads to emergence of Euglenozoa and Chlorarachinoophyte. Red algae also go through similar mechanism of secondary endosymbiosis. The consequent secondary plastid contains four membranes, out of which two are the ones that were inherited from the cyanobacteria during primary endosymbiosis, and the other two are still unknown.

**Figure 2 biomolecules-10-01524-f002:**
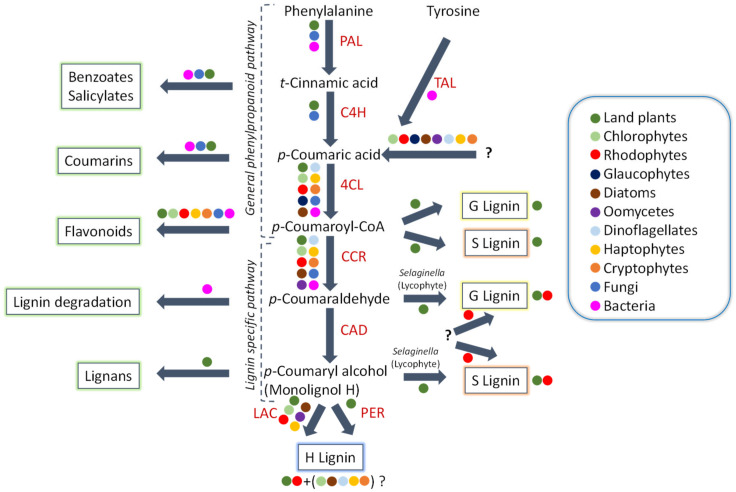
The lignin biosynthesis pathway. Colored dots represent the presence of a given enzyme in a specific taxonomic group. The abbreviations used for the enzyme are: phenylalanine ammonia-lyase; cinnamate 4-hydroxylase (C4H); 4-coumarate:CoA ligase (4CL); cinnamoyl-CoA reductase (CCR); cinnamyl alcohol dehydrogenase (CAD); peroxidase; and laccase (LAC). Question marks (?) signify that the enzyme responsible for the particular function or its substrate is still uncertain, or that the occurrence of a particular compound in a given taxonomic group is suspected but has not been proven. The figure was adapted from Reference [13,49,67].

**Figure 3 biomolecules-10-01524-f003:**
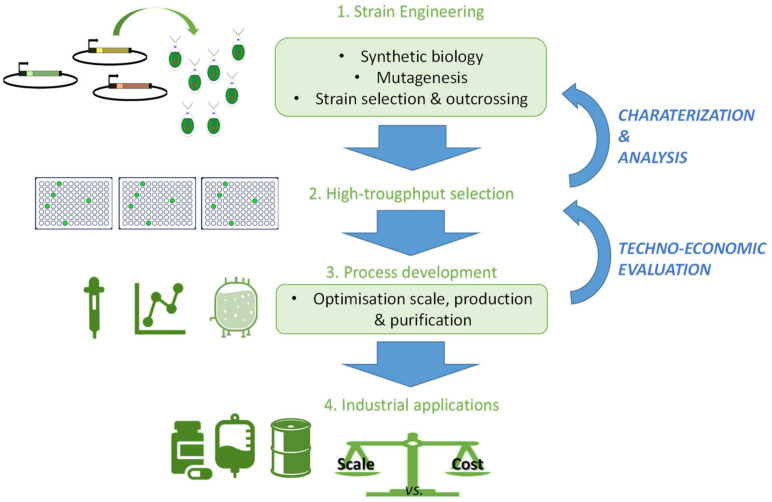
The key steps in bioengineered green algae production (adapted from Scranton et al. [69]).

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
