# Peer review of "An Insight into the Algal Evolution and Genomics"

_biomolecules, 2020, doi:10.3390/biom10111524_

Round 1

Reviewer 1 Report

The article A.K. Khan and co-workers untitled “An insight into the Algal Evolution and Genomics: Consequence for Biomolecules production and Engineering” highlights contribution of algal genomic in the better understanding of algal evolution and adaptation with a biotechnological perspective.
The section concerning the “insights to algal evolution” summarized the understanding evolution of photosynthetic organisms through several endosymbiosis confirmed by recent genomic investigations. It gives an overview of algal evolution accessible and useful to not expert scientist in this field. The second section “Algal genome”, presented some recent results uncovered by genomic studies. Are presented data on diatoms, C. reinhardtii, and comparative genomic which need some clarification on what message the authors wish to transmit.
The two last sections are focused on biotechnological perspectives with at, first, a description of the evolution of the lignin pathway, illustrating how genomic studies allow describing complex pathway. It is not clear however what the real benefit is in term of biotechnology perspectives of understanding the “evolution history” of metabolic pathway. Uncorrelated with previous section, authors presented the recent progress in using microalgae such as C. reinhardtii as “cell factory”.
The links or correlations between the different sections are not obvious as they could be read independently, to my opinion. The idea, the ambition of the article which aims at reviewing how the contribution of a better understanding of algal evolution, the genome sequencing may help to develop new biotechnology, and notably, the production of the novel biomolecules could be improved. More especially, analysis of evolution of lignin pathway presented by the authors which is very interesting, was not correlated to new biotechnological development as it was promised. The algal-based biotechnology presented deals more with the use of microalgae as other expression system such as other known bacterial or yeast systems. The title was very promising but the content was more moderated.

Author Response

AUTHORS: Thank you very much for your comments.

By reading your comments as well as those of reviewer 2, it is clear that we certainly lacked clarity in the explanation of our objectives pursued in the (former) paragraphs 4 and 5. We, thus, have revised our manuscript to reconnect these two incriminated parts to the rest of the manuscript:

1) on the one hand, the part concerning the evolution of phenylpropanoid biosynthesis which had been taken out of paragraph 3 on Algal genome evolution as an illustration of a key innovations in the process of evolution of the aquatic ancestral terrestrial plants adapted to form lignified cell walls. We therefore decide to reconnect this part to the paragraph 3;

2) on the other hand, the part on C. reinhardtii which initially was included to provide basic information to the non-specialist reader on metabolic engineering toolbox provided by this species. By reading the comments of the reviewers, it seems more appropriate to us to include the most appropriate example in the conclusion and future prospect part to support our message and thus reinforced this part as indicated by the reviewers.

Reviewer 2 Report

Sections 1-3 of the manuscript are near-perfect and do not require extensive revisions. They are very interesting and well-written. Figure 1 is excellent in content.

Unfortunately sections 4-6 do not leave up to the expectations set by the  previous sections.

The manuscript title promises the link between the production and engineering of biomolecules in the background of algal genomics and evolution. Whilst the latter is pretty well described the former is severely lacking. The description on biomolecules production is limited exclusively to Phenylpropanoid pathway, this should be expanded significantly by other pathways in the revised manuscript.

The paragraph concerning engineering is also very brief, mostly based on the 2015 paper of Mayfield group with several random references thrown into the mix. This section requires a rewrite to be up to the standard of Biomolecules.

Conclusions section require expanding with new information added in revised sections 4 and 5.

Author Response

AUTHORS: Thank you very much for your comments.

By reading your comments as well as those of reviewer 1, it is clear that we certainly lacked clarity in the explanation of our objectives pursued in the (former) paragraphs 4 and 5. We, thus, have revised our manuscript to reconnect these two incriminated parts to the rest of the manuscript:

1) on the one hand, the part concerning the evolution of phenylpropanoid biosynthesis which had been taken out of paragraph 3 on Algal genome evolution as an illustration of a key innovations in the process of evolution of the aquatic ancestral terrestrial plants adapted to form lignified cell walls. We therefore decide to reconnect this part to the paragraph 3;

2) on the other hand, the part on C. reinhardtii which initially was included to provide basic information to the non-specialist reader on metabolic engineering toolbox provided by this species. By reading the comments of the reviewers, it seems more appropriate to us to include the most appropriate example in the conclusion and future prospect part to support our message and thus reinforced this part as indicated by yourself and the reviewer 1.

Round 2

Reviewer 2 Report

Revised version of the manuscript is better in some aspects ie. the structure is more coherent and it reads better. Since the second part of the manuscript was below the standard of the first, very good part, authors decided to conveniently remove it instead improving the quality of their work.

As a result the revised manuscript stands exactly in the same place as before i.e. needing major revision, what it gained by improved structure, it lost in content and potential impact.

Additional comments: conclusion section is way below the standard expected from the journal.

Figure 2 needs improvement: the ? should be replaced with better desctiption

Figure 3 needs improvement: blue back arrows are pointing towards wrong sections. Results from process development should return to strain engineering not to screening step, or to both. Definetly not to screening only.

Author Response

REVIEWER 2: Revised version of the manuscript is better in some aspects ie. the structure is more coherent and it reads better. Since the second part of the manuscript was below the standard of the first, very good part, authors decided to conveniently remove it instead improving the quality of their work. As a result the revised manuscript stands exactly in the same place as before i.e. needing major revision, what it gained by improved structure, it lost in content and potential impact.

AUTHORS: Thank you for your comment. However this assertion is absolutely false. These two mentioned paragraphs were not removed from the manuscript but included in more appropriate paragraphs in accordance with the first-round comments of both reviewers. This satisfies the Reviewer 1 who’s endorsed this work for publication.

REVIEWER 2: Additional comments: conclusion section is way below the standard expected from the journal.

AUTHORS: Please considered the whole conclusion. Not only the last sentence after the Figure 3 of this work.

REVIEWER 2: Figure 2 needs improvement: the ? should be replaced with better desctiption.

AUTHORS: The answer to the reviewer’s comment about Figure 2 is clearly described in the legend of this figure since the first submitted version of the present work!

REVIEWER 2: Figure 3 needs improvement: blue back arrows are pointing towards wrong sections. Results from process development should return to strain engineering not to screening step, or to both. Definetly not to screening only.

AUTHORS: For Figure 3, as our work is a review, we have redrawn this Figure from an expert article published by a well "recognized" expert in the field published in The Plant Journal (cited in the Figure 3 legend) which is one of the most respected journal in the field with a very high requirement in the peer-review, we can therefore certify that this figure is correct contrary to what this Reviewer claims!